# A Flexible Printed Circuit Board Based Microelectromechanical Field Mill with a Vertical Movement Shutter Driven by an Electrostatic Actuator

**DOI:** 10.3390/s24020439

**Published:** 2024-01-11

**Authors:** Tao Chen, Cyrus Shafai

**Affiliations:** Department of Electrical & Computer Engineering, University of Manitoba, Winnipeg, MB R3T 5V6, Canada; tao.chen@umanitoba.ca

**Keywords:** electric field sensor, electric field mill, flexible PCB, MEMS, micromachining, electrostatic actuator, laser micromachining

## Abstract

Micromachined electric field mills have received much interest for the measurement of DC fields; however, conventional designs with lateral moving shutters could have shutter lifting in the presence of strong fields, which affects their performance. This paper presents a MEMS electric field mill utilizing a vertical movement shutter to address this issue. The sensor is designed and fabricated based on a flexible PCB substrate and is released using a laser-cutting process. The movement of the shutter is driven by an electrostatic actuator. When the driving signal is a sine wave, the shutter moves in the same direction during both the positive and negative half-periods. This facilitates the application of a lock-in amplifier to synchronize with the signal at twice the frequency of the driving signal. In experimental testing, when the vertical shutter is driven at a resonance of 840 Hz, the highest sensitivity of the sensor is achieved and is measured to be 5.1 V/kVm^−1^. The sensor also demonstrates a good linearity of 1.1% for measuring DC electric fields in the range of 1.25 kV/m to 25 kV/m.

## 1. Introduction

A DC (direct current) electric field sensor (EFS) is a device designed to measure and quantify the strength of static or slowly changing electric fields in a given environment. They are useful for applications that involve static or slowly changing electric fields. In power utilities, they can be used to evaluate the electromagnetic environment in an electric power transmission system [1] and the design of insulators [2], and to ensure the safety of power workers [3]. In industrial manufacturing processes, they can be used to prevent electrostatic discharges (ESDs) to protect electronic equipment [4]. In atmospheric science, they are used to study the mechanisms of thunderstorms [5] and predict lightning [6]. They are also useful in the study of climate and geophysics [7].

Common techniques to measure DC electric fields include induction probes, optical sensors, and electric field mills (EFMs). An induction probe has a sensing electrode placed in an electric field, and when it is in equilibrium status, the voltage on the electrode is proportional to the field, which can be measured by using a high-input impedance electrical meter. Some benefits of induction probe type EFSs include a low cost and small size, but they require re-zeroing in shielding conditions, which is inconvenient in long-term applications. Optical sensors are usually based on Pockel’s effect, where an electric field can change a light waveguide crystal’s birefringent properties, and thus, the polarity of the light transversing the light waveguide will be changed [8]. Optical sensors cause minimal distortion of the electric field to be measured, but the optical equipment is often expensive and not stable in a variable temperature environment [9]. An EFM usually has a rotating grounded shutter over a set of sensing electrodes. The rotation of the shutter exposes and covers the sensing electrodes periodically, which generates an AC current with an amplitude proportional to the strength of the electric field. Traditional EFMs can stably work outside in various weather conditions, but they are bulky and require high power consumption and frequent maintenance. Besides these three common measurement techniques, some other types of electric field sensors reported in the literature include a graphene-based device [10], a ferroelectric material-based sensor [11], and a steered-electron sensor [12].

Recently, microelectromechanical field mills (MEFM) have been reported by multiple research groups [13,14,15,16,17,18]. They offer the benefits of small size, low cost, light weight, low power consumption, and minimal maintenance requirements. Similar to a traditional EFM, MEFMs employ grounded shutters vibrating above or adjacent to sensing electrodes. However, when these sensors are exposed to a strong electric field, grounded shutters will lift toward the field, which can affect their shielding ability, significantly reducing the sensor sensitivity. This is because the induced charge on MEFM electrodes is very sensitive to the gap between the shielding shutter and the underlying sensor electrodes. A vertical vibrating shutter type EFM is a promising design to overcome this issue, as the lifting of the shutter can be compensated for by adjusting the vibration range. The concept was first simulated by C. Gong et al. in 2004 [19], where they demonstrated that a vertical movement shutter has a similar shielding effect. However, until now, no MEFM has been reported using a vertical vibration shutter, except S. Ghionea et al., who reported a device that only works for measuring AC fields in 2013 [20].

In this work, we designed a vertical movement shutter type of MEFM by using a flexible PCB (FPCB) substrate. FPCB has the benefit of leveraging a commercialized manufacturing process that is faster and lower cost, making it an ideal choice for academic research in prototyping many MEMS devices without the need for a cleanroom. Recently, an increasing number of devices based on FPCB have been implemented [21,22,23,24]. In comparison to traditional silicon-based devices, their FPCB-based counterparts exhibit comparable performance levels while concurrently showcasing a marked improvement in durability. In this design, the shutter movement is driven by an electrostatic actuator. A simulation was performed to investigate the micro-spring spring constant, resonant frequency, and interference from the driving signal. After the simulation, the sensor was fabricated and tested, and it demonstrated a sensitivity of 5.1 V/kVm^−1^ when operating at resonance.

## 2. Sensor Design

### 2.1. Working Principle

Figure 1 illustrates the working principle of this sensor. It consists of a grounded shutter and grounded sensing electrodes. The shutter is supported by micro-springs, which are not shown in Figure 1. In the center of the shutter, there is a grounded area to form an electrostatic actuator together with an electrode under it. When a voltage is applied to this electrode, the shutter will move. Obviously, if the electrode is closer to the shutter, a stronger electrostatic force can be generated. If the voltage provided is a sine wave *V* = *V*_0_sin(*ωt*), where *ω* = 2*πf*, at *t* = 0 and *t* = *π*, the voltage on the actuator is *V* = 0, and the shutter has no movement (Figure 1a). At *t*
$=\frac{\pi }{2}$, the voltage on the actuator is *V*_0_, and at *t*
$=\frac{3\pi }{2}$, the voltage on the actuator is −*V*_0_. In both situations, the actuator will drive the shutter to move downward for a distance of *d* (Figure 1b). If the shutter moves up and down periodically within an electric field, the sensing electrodes detect variations in the field caused by the changes in the fringing effect. As a result, varying charges are induced, leading to the generation of an alternating current. Based on Gauss’s law, the total induced charge on the sensing electrodes is equal to the total electric flux. At *t* = 0 and *t* = *π*, the electric field on the sensing electrodes is *E*_1_, and the induced charge can be calculated as: (1)$$Q_{1}=\oint_{S}E_{1}\varepsilon_{0}ds$$

At *t*
$=\frac{\pi }{2}$ and *t*
$=\frac{3\pi }{2},$ the shutter movement is *d*, the electric field on the sensing electrodes is *E*_2_, and the induced charge can be calculated as:
(2)$$Q_{2}=\oint_{S}E_{2}\varepsilon_{0}ds$$
where *ε*_0_ ≈ 8.85 × 10^−12^ F/m is the vacuum dielectric constant and *S* is the surface area of the sensing electrodes.

Considering the shutter has the same direction of movement for both the positive and negative half period of the sine wave on the actuator, the induced charge equation can be written as:(3)$$∆Q=\left|\left(Q_{1}-Q_{2}\right)\sin\left(\omega t\right)\right|$$

This is equivalent to $\left(Q_{1}-Q_{2}\right)\sin\left(\omega t\right)$ modulated with an amplitude 1 square wave at the same frequency. Substituting the Fourier series of the ideal square wave into (3), then:(4)$$∆Q=\frac{4}{\pi }\left(Q_{1}-Q_{2}\right)\sin\left(\omega t\right)\sum_{n=1}^{\infty }\frac{1}{2n}sin\left(\left(2n-1\right)\omega t\right)$$

Expanding Equation (4) we obtain:$$∆Q=\frac{4}{\pi }\left(Q_{1}-Q_{2}\right)\sin\left(\omega t\right)\left[\sin\left(\omega t\right)+\frac{1}{3}\sin\left(3\omega t\right)+\frac{1}{5}\sin\left(5\omega t\right)+\cdots \right]$$
$$∆Q=\frac{4}{\pi }\left(Q_{1}-Q_{2}\right)\left[\sin\left(\omega t\right)\sin\left(\omega t\right)+\frac{1}{3}\sin\left(\omega t\right)\sin\left(3\omega t\right)+\frac{1}{5}\sin\left(\omega t\right)\sin\left(5\omega t\right)+\cdots \right]$$
(5)$$∆Q=\frac{4}{\pi }\left(Q_{1}-Q_{2}\right)\left\{\frac{1}{2}(1-cos(2\omega t))-\frac{1}{6}\left[\cos\left(4\omega t\right)-\cos(2\omega t)\right]-\frac{1}{10}\left[\cos\left(6\omega t\right)-\cos(4\omega t)\right]+\cdots \right\}$$

Ignoring all high frequency components in Equation (5), and substituting (1) and (2) into (5), if only considering components at frequency 2*ω*, Equation (5) can be simplified as:(6)$${∆Q}_{2\omega }=-\frac{4\cos\left(2\omega t\right)}{3\pi }\oint_{S}{(E}_{1}-E_{2})\varepsilon_{0}ds$$

And so, the generated current at frequency 2*ω* is:(7)$$I_{2\omega }=\frac{{d∆Q}_{2\omega }}{dt}$$

Substituting (6) into (7), we get the equation for the current at frequency 2*ω*:(8)$$I_{2\omega }=\frac{8\omega \sin\left(2\omega t\right)}{3\pi }\oint_{S}(E_{1}-E_{2})\varepsilon_{0}ds$$

This current signal is detectable after being amplified using a high impedance amplifier and then extracted from the noise by using a lock-in amplifier.

### 2.2. Working Principle

COMSOL Multiphysics software 6.0 was used for the simulations. The COMSOL material library was employed for defining the material properties of copper and polyimide. The 3D model created in COMSOL is shown in Figure 2a, and its dimension parameters are listed in Table 1. Figure 2b illustrates the schematic diagram of the model, demonstrating the arrangement of the sensor components. As we can see, the sensing electrodes and shutter are on one polyimide substrate, while the driving electrode, guard line, and bottom shielding electrode are on another polyimide substrate. In practice, they can be separated by using a spacer, but in the simulation, a spacer is not used in the model. The grounded shutter is supported by four S-shaped micro-springs and is placed 100 µm over the driving electrode. The sensing electrodes are at the same height as the shutter. The first simulation is to find the spring constant and shutter movement distance driven by the actuator. Figure 3a shows the shutter’s downward movement when a voltage of 200 V is applied on the actuator, which results in a peak deflection of 5.59 µm. Cross-section views of the shutter downward movement are shown in Figure 3b–d. Figure 4 plots the simulated deflection for the drive voltages of 50 V, 100 V, 150 V, and 200 V, resulting in electrostatic forces on the shutter of 4.82 × 10^−6^ N, 1.97 × 10^−5^ N, 4.6 × 10^−5^ N, and 8.69 × 10^−5^ N, respectively. The overall spring constant is calculated to be 15.5 N/m. 

The resonant frequency of a micro-springs supported shutter is also simulated. It has multiple orders of vibration modes, and Table 2 lists the results of the first three modes. As we can see, the first order is up and down at 493 Hz, the second order is tilt along the x-axis at 568 Hz, and the third order is tilt along the y-axis at 1045 Hz. Both the secondary and third orders have half of the structure move up and half move down, where the induced charges on each side will cancel each other; therefore, no signal will be generated. The first-order vibration mode will have a maximized induced charge.

The high voltage on the driving electrode is a concern in that it may produce extra induced charges on the sensing electrodes. To minimize this interference, in the design, two grounded guard lines are placed on each side of the driving electrode to separate it from the sensing electrode, and another grounded electrode is placed under the driving electrode and guard lines to prevent an electric field from emanating from below (on the other side of the PCB). Figure 5 depicts the distribution of the electric field between the driving electrode and the surrounding grounded structures. We can see that the sensing electrodes are minimally affected by the driving voltage. For example, when 200 V at 246.5 Hz is applied to the driving electrode (the shutter has a resonant frequency of 493 Hz), the induced charge on the sensing electrodes is 2.5 fC. The generated current is:(9)$$I_{\omega }=\frac{d∆Q}{dt}=\omega \left(Q_{1}-Q_{2}\right)\cos\left(\omega t\right)$$

Using Equation (9), the amplitude of the current is calculated to be 3.9 pA. In an electric field of 10 kV/m, the movement of the shutter (driven by 200 V) results in an induced charge of 0.17 pC, while at rest (0 V drive), the induced charge is 0.153 pC. Using Equation (8), the amplitude of the generated current is calculated to be 44.7 pA. Compared to the interference from the driving electrode, the sensing signal is 11.5 times stronger than the driving signal interference, and the frequency is two times higher.

## 3. Sensor Fabrication

After the simulation, the sensor patterns were designed using a free and open-source PCB design tool named KiCAD 7.0 The generated Gerber and drill files are demonstrated in Figure 6a. These files were sent to a PCB manufacturer called PCBway (https://www.pcbway.com/, accessed on 30 November 2023) to fabricate the structure. The flexible PCB samples are shown in Figure 6b,c, where Figure 6b is the shutter and sensing electrodes, and Figure 6c is the actuator driving electrode. These samples have two conductive layers, where Figure 6b,c shows the front view, and the black features on these two pictures are the features on the back side. According to the manufacturer’s datasheet, the thickness of the copper is 18 µm and that of the polyimide is 25 µm, and minimum line width is 60 µm. 

The shutter sample needs to be released to separate the sensing electrodes. This was completed by using a laser etching process. The equipment used is the A Series Laser Micromachining system, which is an ultraviolet (355 nm) diode-pumped solid-state picosecond laser dicer from Oxford Lasers Ltd. (Didcot, Oxfordshire, UK). The cutting paths were created by using AlphaCAM 2023 R2 software. The laser beam peak pulse energy was set to 0.12 mJ and the pulse duration was 6 ps. The laser pulse frequency setting was 400 Hz, cutting speed 1 mm/s, and the diameter of the laser beam was 10 µm. Using 30% of this laser power, 15 cutting passes were required to cut through the polyimide-only areas and 30 passes were required for areas having both copper and polyimide. 

The released micro-springs, shutter, and sensing electrodes are shown in Figure 6e, where two corners are cut to expose the electrodes on the layer below it. After soldering wires to the sample, then the sensor is assembled by aligning the released sample on top of the actuator driving electrode and using taps to fix all of the parts. As the polymer substrate is partially transparent, in the alignment process, the cross-shaped alignment marks on both layers will make the process easier.

Figure 7a shows the picture of the released micro-springs, shutter, and sensing electrodes. The picture was taken using a Leica DM6 M microscope (Leica Microsystems, Wetzlar, Germany). As we can see, the micro-springs, shutter, and sensing electrodes have slight deformations, but the shutter and the sensing electrodes are still aligned very well. After assembly, the separation distance from the shutter to the driving electrode under it is measured to be 105 µm by using a microscope. Figure 7b shows the initial location of the shutter fingers and sensing electrodes in the zoomed-in area of Figure 7a. As an illustration of the shutter movement test, the application of 200 V to the driving electrode results in noticeable shutter finger movement, shown in Figure 7c. We can see that the sensing electrodes remain focused and unchanged, while the shutter fingers move downward, altering the focal status. By refocusing the shutter finger and observing the displacement of the fine object adjustment knob of the microscope, the movement of the shutter fingers was measured as 10 µm. As detailed in Section 2.2, the electrostatic force exerted on the shutter at 200 V is 8.69 × 10^−5^ N. Therefore, the spring constant of the shutter’s supporting springs is calculated to be 8.7 N/m.

## 4. Sensor Testing

### 4.1. Test Setup

The fabricated sensor was tested in the laboratory. Figure 8 shows a functional diagram of the test setup. The sensor was placed on a grounded metal plate, and the distance from the sensor surface to the metal plate was measured to be 0.5 mm. Another metal plate was placed above the sensor, and the two metal plates were separated by two 4.5 mm thick spacers on each side, then the distance from the top metal plate to the surface of the sensor was 4 mm. A DC power supply was connected to the top metal plate to generate a test DC electric field. The actuator driving sine wave was generated by an Agilent 33120A signal generator (Agilent Technologies, Santa Clara, CA, USA), followed by three 1:7 transformers in parallel in the primary side and in series in secondary side. When an amplitude 10 V sine wave was provided by the signal generator, the output driving signal amplitude was measured to be 193 V. The output of the sensing electrodes was sent to a transimpedance amplifier (TIA), with a gain of 10^7^. After the TIA, an interfering signal was detected, which had the same frequency as the actuator driving signal and overloaded the lock-in amplifier. This is because the deformation of the shutter after release and the flatness of the electrodes was not perfect, which caused the electric field from the driving electrode to reach the sensing electrodes more easily. 

To minimize this interference while not attenuating the sensing signal, a sixth-order high-pass Butterworth filter with a cutoff frequency of 500 Hz was employed. The gain for infinite frequency was set to 1, and the quality factor Q was selected to be 0.707. The frequency response of this filter was simulated by using Cadence Orcad 22.1 Pspice (Cadence, San Jose, CA, USA), and Figure 9 shows the result. Since the driving signal frequency was only half the sensing frequency, it was attenuated more. After the high-pass filter, the signal was fed to an SR510 lock-in amplifier. During the test, the lock-in amplifier integration time was set to be 1 s, and the lock-in frequency was set to be 2*f*. The output of the lock-in amplifier was sent to an oscilloscope and a digital multimeter. By adjusting the lock-in amplifier sensitivity, different ranges of electric field can be measured.

### 4.2. Frequency Response Test

After setting up the test apparatus, the first test performed was to find the actual resonant frequency, where the sensor has its maximum response. In this test, we selected the lock-in amplifier sensitivity to be 2 mV. Then we set the signal generator amplitude to be 8 V. After transformer amplification, the amplitude of the signal driving the actuator was 153 V. The frequency was swept from 160 Hz to 580 Hz with an increment of 10 Hz. The DC power supply output voltage was set to be 100 V, which can create a test electric field of 25 kV/m at each frequency, enabling and disabling the output of the DC power supply. The responses of the sensor shown on the digital multimeter were recorded and are plotted in Figure 10. The analog output of the SR510 lock-in amplifier is given by the equation:(10)$$V_{out}=10A_{e}(A_{v}V_{i}\cos\emptyset +V_{OS})$$
where *A_e_* is the expanded setting on the lock-in amplifier panel; *A_v_* equals the reciprocal of the sensitivity setting, which is 2 mV in this test; *V_i_* is the magnitude of the input signal to the lock-in amplifier; *Φ* is the phase difference between the signal and reference; and *V_OS_* is the offset. However, the SR510 only gives the in-phase component *X* of *V_out_*. The amplitude of *V_out_* can be derived by both the in-phase and quadrature component *Y*:(11)$$Z=\sqrt{X^{2}+Y^{2}}$$

In this test, for each frequency, we measured the *X* component by setting the phase to be 0°, and measured the Y component by setting the phase to be 90°, then calculated the Z component using Equation (11). The test results are shown in Figure 10. The *Z* component maximum occurs at 420 Hz, which indicates that the shutter resonant frequency is 840 Hz.

### 4.3. Sensor Sensitivity Test

In this section, we explore the sensor sensitivity. For a linear sensor, the transfer function can be described by:(12)E=A+Bs
where *s* is the input signal, *E* is the output signal, *A* is the output signal *E* at zero input signal *s* = 0, and *B* is the slope of the line. *B* is also called sensitivity [25]. Therefore, the sensor sensitivity is:(13)$$B=\frac{E-A}{s}$$

For this test, we set the signal generator amplitude to be 10 V, and after transformer amplification, the amplitude of the signal driving the actuator was 193 V. As the in-phase *X* component peaked at 800 Hz (the driving signal at 400 Hz), we chose the lock-in amplifier to display the *X* component, the phase was set to be 0°, and we set the signal generator output frequency to be 400 Hz. The DC power supply output voltage was set to be 10 V, which provided a 2.5 kV/m test electric field when turned on. To explore the highest sensitivity of the sensor, the SR510 lock-in amplifier sensitivity was pushed to the lowest before overload to be 100 µV. Figure 11 shows the test results. As we can see, when the 2.5 kV/m field turns on and off, the sensor output is averaged to be 12.8 V. The sensor sensitivity can be calculated to be 5.1 V/kVm^−1^. The noise level of the sensor is 0.3 V, and the resolution of the sensor is calculated to be 62.5 V/m. Table 3 compares the EFMs presented in this paper and some reported in recent years. This sensor demonstrates significantly higher sensitivity and comparable resolution.

### 4.4. Sensor Response Linearity Test

In order to explore the sensor response linearity, a test was performed to measure the sensor response from 1.25 kV/m to 25 kV/m. In this test, the driving signal amplitude was 193 V, and the frequency 400 Hz. The lock-in amplifier sensitivity was selected to be 1 mV. When the DC power supply swept the voltage from 5 V to 100 V, electric fields from 1.25 kV/m to 25 kV/m were generated. Figure 12 plots the sensor response (each reading has an error of ±0.03 V). The linearity of the sensor is calculated to be 1.1%, with an average sensitivity of 0.49 V/kVm^−1^.

## 5. Conclusions

This paper introduces an MEFM employing a vertical movement shutter based on a flexible PCB substrate. The shutter is driven by an electrostatic actuator. The sensor structure was fabricated by a commercial PCB manufacturing process followed by laser cutting for release. The fabricated sensor demonstrated a linear response in electric fields ranging from 1.25 kV/m to 25 kV/m. The highest sensitivity was measured to be 5.1 V/kVm^−1^. This sensor has the potential to compensate for the shutter lift-up issue of common MEFMs by providing an initial bias voltage on the actuator. This allows this type of sensor to be applied in a wide range of DC electric field measurements from sub-kV/m to over MV/m.

## Figures and Tables

**Figure 1 sensors-24-00439-f001:**
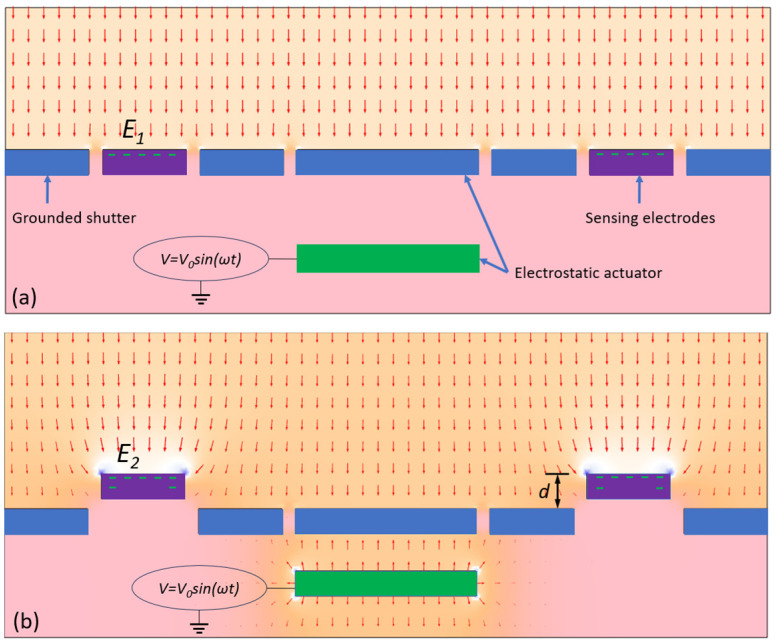
Sensor working principle. When an AC voltage is applied to the electrostatic actuator: (**a**) At t=0 and t=π, the shutter is at the same height as the sensing electrodes. (**b**) At $t=\frac{\pi }{2}$ and $t=\frac{3\pi }{2}$, the shutter is pulled down and the sensing electrodes are exposed to a stronger field and more surface area is exposed.

**Figure 2 sensors-24-00439-f002:**
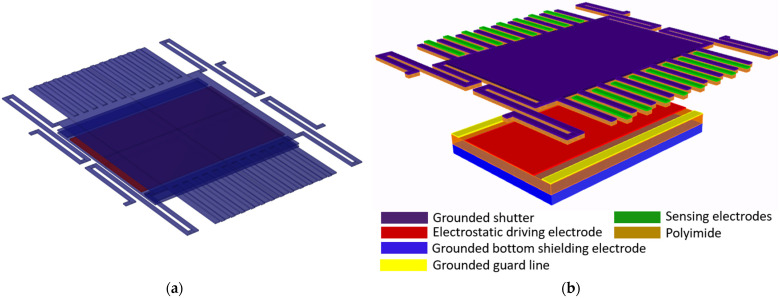
Sensor 3D model. (**a**) COMSOL 3D model. (**b**) 3D model schematic diagram.

**Figure 3 sensors-24-00439-f003:**
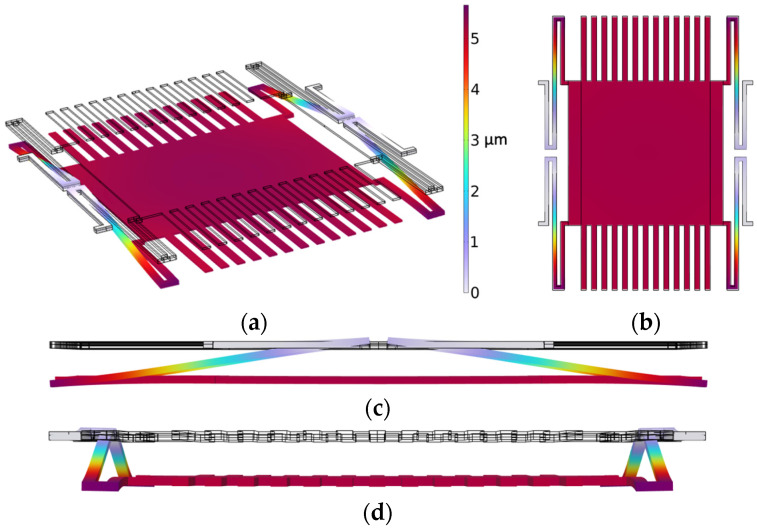
Shutter movement simulation. (**a**) Shutter movement under force of 86.9 µN. (**b**) x-y plane view. (**c**) y-z plane view. (**d**) x-z plane view.

**Figure 4 sensors-24-00439-f004:**
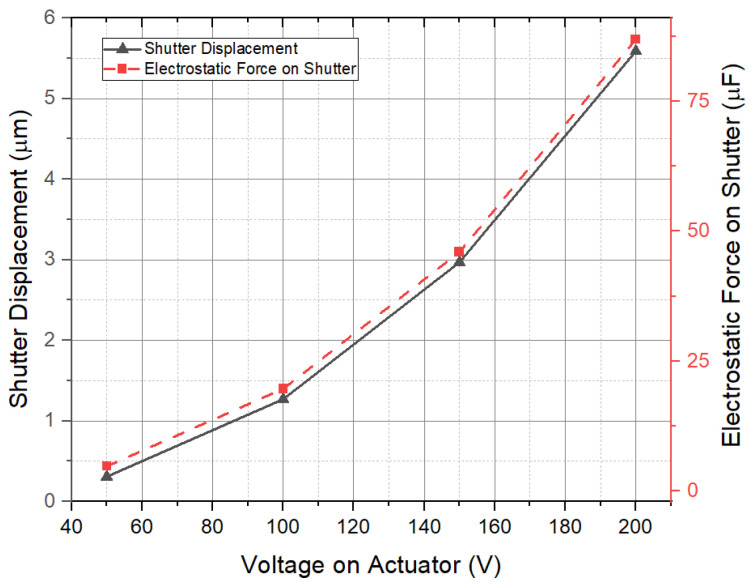
Simulation results for the shutter displacement and electrostatic force on the shutter when driven by the actuator.

**Figure 5 sensors-24-00439-f005:**
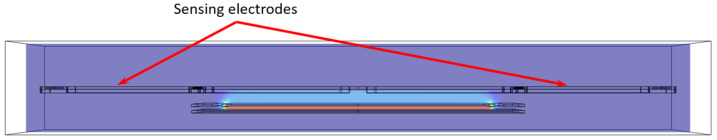
Driving electric field distribution.

**Figure 6 sensors-24-00439-f006:**
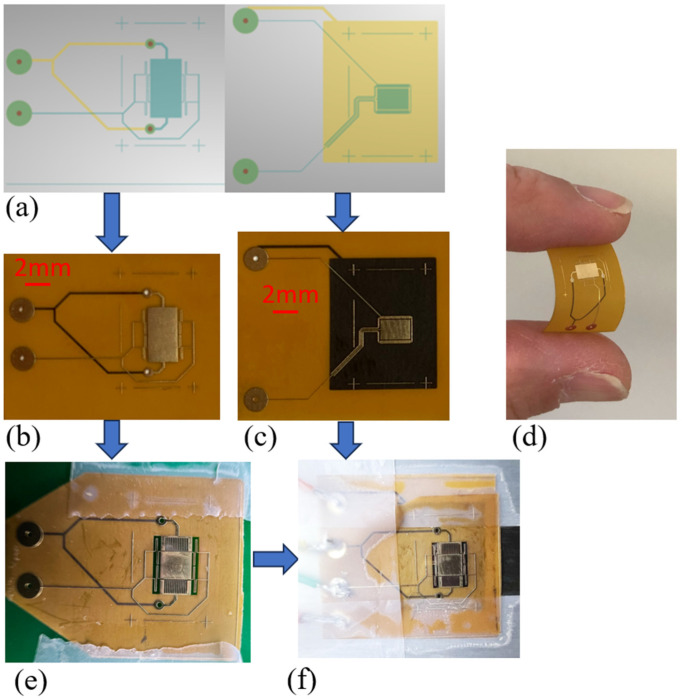
Sensor fabrication process. (**a**) The view of the design PCB and generate Gerber files. (**b**) Manufactured shutter and sensing electrodes. (**c**) Manufactured actuator driving electrodes. (**d**) Bending the sample manually to show the flexibility. (**e**) After laser cutting, the micro-springs are released and the shutter and sensing electrodes are separated. (**f**) Place (**e**) on top of (**c**) to form the sensor.

**Figure 7 sensors-24-00439-f007:**
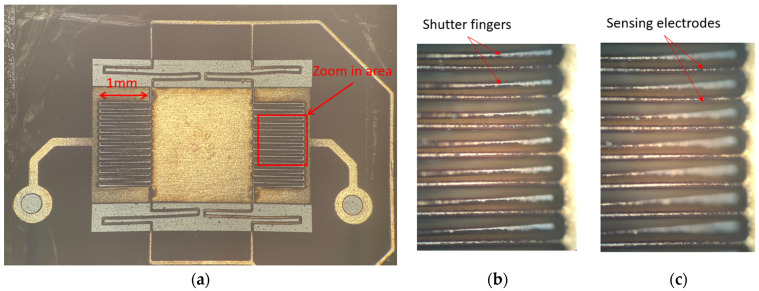
Released shutter and sensing electrodes under the microscope. (**a**) Whole area. (**b**) Zoom picture of the electrodes at the initial rest position. (**c**) Zoom picture after applying 200 V on the driving electrode.

**Figure 8 sensors-24-00439-f008:**
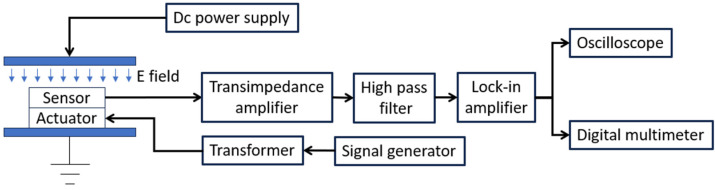
Test function diagram.

**Figure 9 sensors-24-00439-f009:**
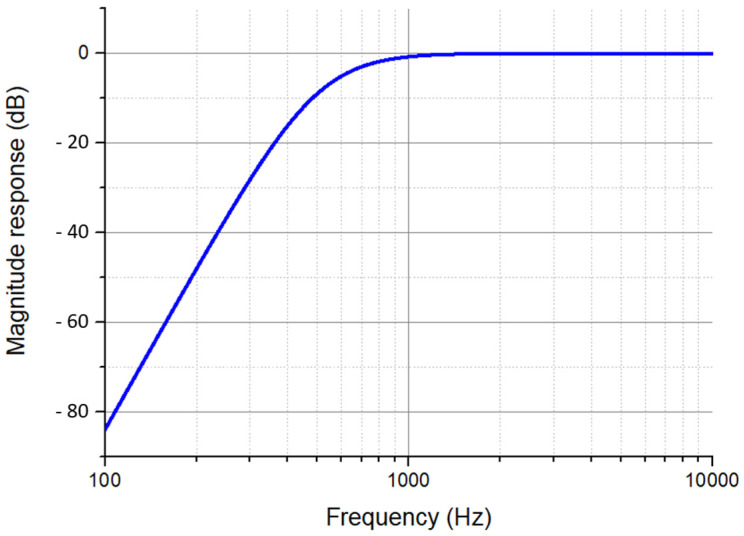
High pass filter frequency response simulation.

**Figure 10 sensors-24-00439-f010:**
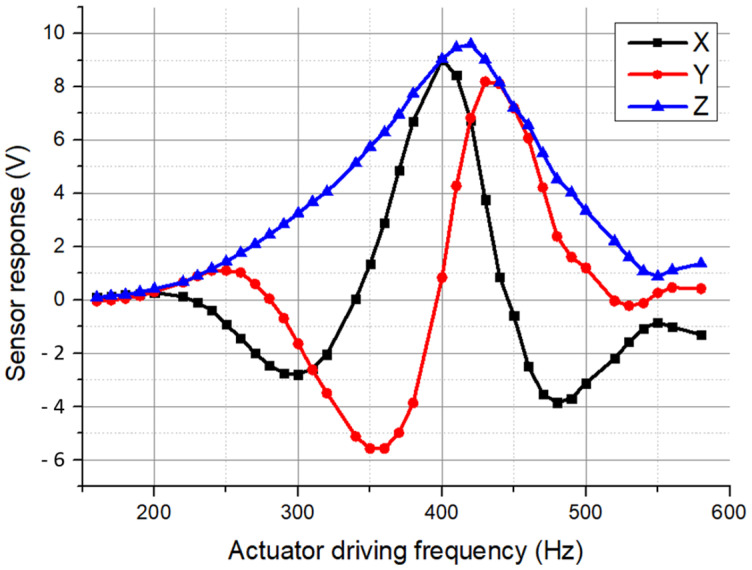
Sensor frequency response test results.

**Figure 11 sensors-24-00439-f011:**
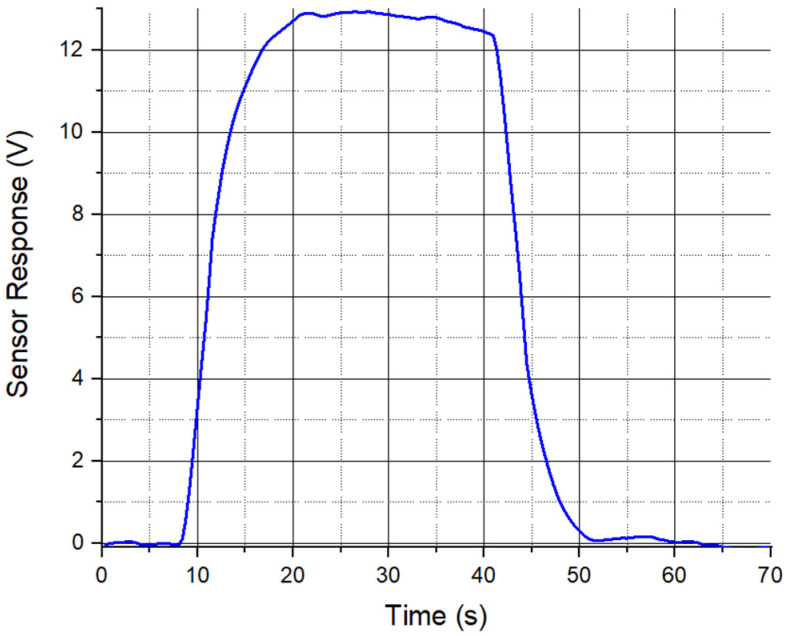
Sensor sensitivity test results.

**Figure 12 sensors-24-00439-f012:**
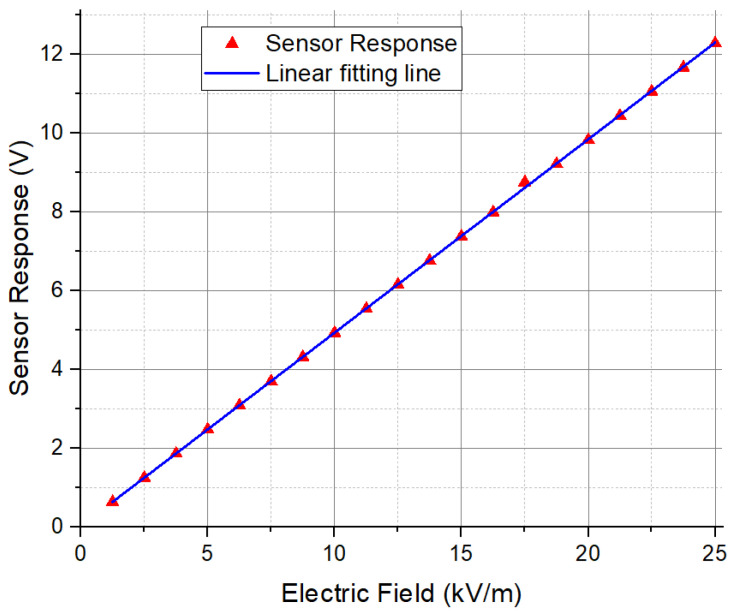
Sensor response to DC electric fields from 1.25 kV/m to 25 kV/m.

**Table 1 sensors-24-00439-t001:** COMSOL 3D model properties.

Property	Value (µm)
Spring length	2000
Spring width	60
Copper thickness	18
Polyimide thickness	25
Sensing electrodes length	1000
Sensing electrodes width	60
Shutter finger length × width	1000 × 60
Gap between sensing electrodes and shutter finger	10
Electrostatic driving electrode width	2120
Electrostatic driving length	1920
Guard line length × width	1920 × 120
Distance from electrostatic driving electrode to shutter	100

**Table 2 sensors-24-00439-t002:** Resonant frequency simulation results.

Mode	Picture	Frequency (Hz)
1	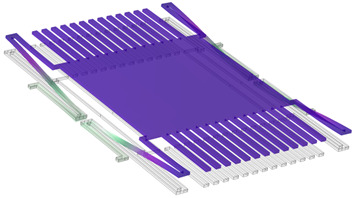	493
2	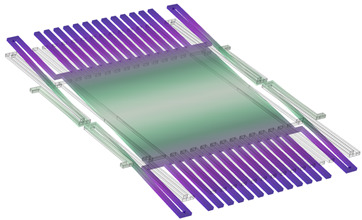	568
3	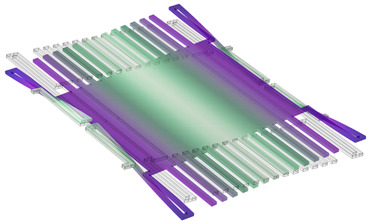	1045

**Table 3 sensors-24-00439-t003:** Comparison between references and this work.

Reference	Year	Sensitivity	Resolution
MEMS EFM [13]	2001	40 µV (kV/m)^−1^	N/A
Thermal actuator EFM [14]	2009	0.1 mV (kV/m)^−1^	42 V/m
Thermal actuator EFM [15]	2006	0.4 mV (kV/m)^−1^	240 V/m
Torsional Resonance EFM [16]	2018	5 mV (kV/m)^−1^	N/A
Comb drive EFM [18]	2017	10 mV (kV/m)^−1^	N/A
EFM in this paper	2023	5.1 V (kV/m)^−1^	62.5 V/m

## Data Availability

Data sharing does not apply to this article as no new data were created or analyzed in this study.

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
