# Peer review of "A Flexible Printed Circuit Board Based Microelectromechanical Field Mill with a Vertical Movement Shutter Driven by an Electrostatic Actuator"

_sensors, 2024, doi:10.3390/s24020439_

Round 1

Reviewer 1 Report

Comments and Suggestions for Authors

This paper presents a flexible PCB-based MEMS field mill with a vertical movement shutter driven by an electrostatic actuator. The sensitivity achieved by the device is better than the reported works. I have the following suggestions for the authors.

1. Add the color-coded label to the results of the COMSOL simulation in Fig. 3.

2. I recommend adding a scale bar to the images in Fig. 6.

3. Add a unit to the y-axis label in Fig. 10. 

4. How about the power consumption of the devices reported in table 3? 

Author Response

Dear reviewer, 

   We would like to thank you for providing valuable feedback on our manuscript. We have carefully considered your comments and suggestions and revised the manuscript accordingly.  Our responses to the comments and suggestions are presented as follows:

1. Add the color-coded label to the results of the COMSOL simulation in Fig. 3. 
Thanks for your suggestion. Color-coded label has been added to Figure 3 (a).
2. I recommend adding a scale bar to the images in Fig. 6. 
Thanks for your suggestion. A scale bar has been added to Figure 6.
3. Add a unit to the y-axis label in Fig. 10.  
Thanks for your suggestion. Figure 10 is now Figure 9, and we replot the frequency response using magnitude response (dB) as y-axis.
4. How about the power consumption of the devices reported in table 3?  
The information about the power consumption of the devices in table 3 are not provided in those papers, so we are unable to add it. 

Best regards,

Reviewer 2 Report

Comments and Suggestions for Authors

The author proposed a novel approach to micromachined electric field mills based on micro-electromechanical system (MEMS), a key technology for measuring direct current (dc) fields. The proposed MEMS electric field mill employs a vertical movement shutter. The design and fabrication of MEMS on a flexible printed wire board (PCB) substrate are demonstrated in this work. Therefore, I recommend that the manuscript be accepted after minor revision. My comments are listed as follows:     

1.      The schematic drawing of Figure 2 is not clear enough; I suggest the author use 3D design software and then render each part or component using a different color. It is better to give a zoom-in picture of each part or component.

2.      The cross-section of the x-plane view, y-plane view, and z-plane view should be added in Figure 3. And give the coordinate system clearly.  

3.      The shutter displacement of each axis should be present in Figure 4.

4.      The author claims that they fabricate MEMS in a flexible PCB. However, the flexibility of PCB is not shown in Figure 6 and Figure 7. In addition, this work does not present the bend or twist test for the proposed MEMS electric field mill.

5.      The top or side view pictures of the shutter's initial condition and displacement condition should be present in Figure 7 to improve the discussion.

6.      Figure 9 looks messy. I suggest the author present the schematic drawing of the experiment setup instead of the whole real picture of the experiment setup

7.      The equation or the definition of the sensor's sensitivity is not clearly presented and discussed in this manuscript.

Author Response

Dear reviewer,

   We would like to thank you for providing valuable feedback on our manuscript. We have carefully considered your comments and suggestions and revised the manuscript accordingly. Changes in the revised manuscript are highlighted with red colored font. Our responses to the comments and suggestions are presented as follows:

1.      The schematic drawing of Figure 2 is not clear enough; I suggest the author use 3D design software and then render each part or component using a different color. It is better to give a zoom-in picture of each part or component. 
Thanks for your suggestion. We added Figure 2 (b), a schematic diagram of the 3D model, to provide a clear illustration of the assembly of model parts.
2.      The cross-section of the x-plane view, y-plane view, and z-plane view should be added in Figure 3. And give the coordinate system clearly.   
Thanks for your suggestion. xy-plane view, x-z plane view, and y-z plane view are added as Figure 3 (b), (c), and (d).
3.      The shutter displacement of each axis should be present in Figure 4. 
Thanks for your suggestion. But the shutter operates through an electrode positioned beneath it, primarily exhibiting displacement in the z-direction, as confirmed through simulation. Although there is minimal x and y displacement in real-world conditions, these deviations are negligible when compared to the predominant z-direction displacement. Therefore, Figure 4 only plot z direction displacement.

4.      The author claims that they fabricate MEMS in a flexible PCB. However, the flexibility of PCB is not shown in Figure 6 and Figure 7. In addition, this work does not present the bend or twist test for the proposed MEMS electric field mill. 
Thanks for your suggestion. This field mill is designed and fabricated based on flexible PCB, where the substrate (polyimide) itself is flexible. We added a picture in Figure 6 (d) to show the flexibility of the sample. While the substrate is flexible, the sensor is designed to operate at z axis deflection. And so study of operation with bend or twist condition is beyond the scope of this work. 
5.      The top or side view pictures of the shutter's initial condition and displacement condition should be present in Figure 7 to improve the discussion. 
Thanks for your suggestion. We add Figure 7 (b), (c) to show the original and relocated positions of the shutter in the zoomed-in area of Figure 7(a).
6.      Figure 9 looks messy. I suggest the author present the schematic drawing of the experiment setup instead of the whole real picture of the experiment setup 
Thanks for your suggestion. As the experimental setup schematic is already illustrated in Figure 8, Figure 9 has been omitted. 
 7.      The equation or the definition of the sensor's sensitivity is not clearly presented and discussed in this manuscript. 
Thanks for your suggestion. we add the introduction of sensitivity in section 4.3.

Best regards,

Reviewer 3 Report

Comments and Suggestions for Authors

I would like to see all the dimensions of the sensor, specifically in figure 6.

Comments on the Quality of English Language

Slightly awkward English, missing some articles ("a" and "the") in places.  Perfectly easy to understand.  On line 47 "filed" should be "field".  Line 91 "hight" should be "height"

Author Response

Dear reviewer,

We would like to thank you for providing valuable feedback on our manuscript. We have carefully considered your comments and suggestions and revised the manuscript accordingly. Changes in the revised manuscript are highlighted with red colored font. Our responses to the comments and suggestions are presented as follows:

I would like to see all the dimensions of the sensor, specifically in figure 6. 

Thanks for your suggestion, we add a scale bar in Figure 6.

Comments on the Quality of English Language 

Slightly awkward English, missing some articles ("a" and "the") in places.  Perfectly easy to understand.  On line 47 "filed" should be "field".  Line 91 "hight" should be "height" 

Thanks for your suggestion. we have fixed the error and checked the paper grammar again.

Best regards

Round 2

Reviewer 1 Report

Comments and Suggestions for Authors

The manuscript has been revised and improved.

Reviewer 2 Report

Comments and Suggestions for Authors

The author has excellent responses to the reviewer’s comments. I appreciated the authors’ effort in modifying their manuscript. Therefore, I would like to accept this manuscript. 

Reviewer 3 Report

Comments and Suggestions for Authors

No further comments from initial review